# Development of algorithms for identifying patients with Crohn's disease in the Japanese health insurance claims database

Hiromu Morikubo[1,2,3], Taku Kobayashi[1]*, Tomohiro Fukuda[1,2], Takayoshi Nagahama[4], Tadakazu Hisamatsu[3], Toshifumi Hibi[1]

1 Center for Advanced IBD Research and Treatment, Kitasato University Kitasato Institute Hospital, Minato-ku, Tokyo, Japan, 2 Department of Gastroenterology and Hepatology, Kitasato University Kitasato Institute Hospital, Minato-ku, Tokyo, Japan, 3 Department of Gastroenterology and Hepatology, Kyorin University School of Medicine, Mitaka-shi, Tokyo, Japan, 4 Data Innovation Lab, Japan Medical Data Center Co., Ltd., Minato-ku, Tokyo, Japan

* kobataku@insti.kitasato-u.ac.jp

## Abstract

**Data Availability Statement:** All files are available from the GitHub database (https://github.com/HiromuMorikubo/pone2021).

### Background

Real-world big data studies using health insurance claims databases require extraction algorithms to accurately identify target population and outcome. However, no algorithm for Crohn's disease (CD) has yet been validated. In this study we aim to develop an algorithm for identifying CD using the claims data of the insurance system.

### Methods

A single-center retrospective study to develop a CD extraction algorithm from insurance claims data was conducted. Patients visiting the Kitasato University Kitasato Institute Hospital between January 2015–February 2019 were enrolled, and data were extracted according to inclusion criteria combining the Tenth Revision of the International Statistical Classification of Diseases and Related Health Problems (ICD-10) diagnosis codes with or without prescription or surgical codes. Hundred cases that met each inclusion criterion were randomly sampled and positive predictive values (PPVs) were calculated according to the diagnosis in the medical chart. Of all cases, 20% were reviewed in duplicate, and the inter-observer agreement (Kappa) was also calculated.

### Results

From the 82,898 enrolled, 255 cases were extracted by diagnosis code alone, 197 by the combination of diagnosis and prescription codes, and 197 by the combination of diagnosis codes and prescription or surgical codes. The PPV for confirmed CD cases was 83% by diagnosis codes alone, but improved to 97% by combining with prescription codes. The inter-observer agreement was 0.9903.

**Funding:** This study was funded by JMDC Inc. The funder provided support in the form of salaries for TN, but did not have any additional role in the study design, data collection and analysis, decision to publish, or preparation of the manuscript. There was no additional external funding received for this study. The specific roles of these authors are articulated in the author contributions section.

**Competing interests:** HM has received research grants from Japan Foundation for Applied Enzymology. TK has served as a speaker, a consultant or an advisory board member for Abbvie, Alfresa Pharma, Janssen Pharma, Takeda, Mitsubishi Tanabe Pharma, Pfizer, Mochida, and received research grants from Nippon Kayaku, EA Pharma, Otsuka Holdings, JIMRO, Abbie, Zeria. FT has received research grants from Mitsubishi Tanabe Pharma. TN are employees of JMDC Co. Ltd., holds shares in JMDC Co. Ltd. TaH has served as a speaker, a consultant or an advisory board member for Mitsubishi Tanabe Pharma, AbbVie GK, EA Pharma, Kyorin Pharma, JIMRO, Janssen Pharmaceutical, Mochida Pharmaceutical, Takeda Pharmaceutical, and received research grants from Alfresa Pharma, EA Pharma, Mitsubishi Tanabe Pharma, AbbVie GK, JIMRO, Zeria Pharmaceutical, Daiichi-Sankyo, Kyorin Pharmaceutical, Nippon Kayaku, Astellas Pharma, Takeda Pharmaceutical, Pfizer, Mochida Pharmaceutical. ToH has served as a speaker, a consultant or an advisory board member for Aspen Japan, Abbvie GK, Ferring, Gilead Sciences, Janssen, JIMRO, Mitsubishi Tanabe Pharma, Mochida Pharmaceutical, Nippon Kayaku, Pfizer, Takeda Pharmaceutical, Zeria, and received research grants from Abbvie, EA Pharma, JIMRO, Otsuka Holdings, Zeria, and received scholarship grants from Zeria. This does not alter our adherence to PLOS ONE policies on sharing data and materials.

## Conclusions

Single ICD-code alone was insufficient to define CD; however, the algorithm that combined diagnosis codes with prescription codes indicated a sufficiently high PPV and will enable outcome-based research on CD using the Japanese claims database.

## Introduction

Crohn's disease (CD) is a chronic inflammatory bowel disease (IBD) of unknown etiology [1]. Recent progress on treatment for IBD has been remarkable, and many new drugs have been launched following randomized control trials (RCTs) [2]. At the same time, multiple clinical questions have arisen to help adapt the increased treatment options to better suit patients' needs in clinical practice. Consequently, the importance of observational studies, as well as RCTs, are being reevaluated [3]. In fact, it has been demonstrated that RCTs represent only a small proportion of patients with IBD in real-world practice [4]. In this respect, large-scale observational studies are also needed.

The incidence and prevalence of CD are higher in Western countries [5] and are also increasing in Asian countries, including Japan [6]. Its prevalence is 1.51-322/100,000 in Western countries [5] and 55.6/100,000 in Japan [7]. When conducting real-world observational studies requiring a large number of patients in Japan, it is often difficult to obtain a sufficient sample size from a single or small number of institutions. For diseases with low prevalence, the claims database can therefore be a useful tool for conducting large-scale real-world observational studies [8, 9]. In fact, various epidemiological studies using the claims database have been successfully conducted [10–12], and the usefulness of these databases has also been proven in IBD [8, 13–15]. However, it is important to note that the diagnosis in the claims database may not always reflect the final medical diagnosis made in clinical practice, and validation studies are therefore necessary for each disease [16–19]. Furthermore, in Japan, the validity of the Tenth Revision of the International Statistical Classification of Diseases and Related Health Problems (ICD-10) codes registered in the claims data for CD has not been evaluated. Thus, the reliability of claims database studies on CD using ICD-10 codes has not yet been confirmed. Therefore, the purpose of this study was to develop an algorithm for identifying CD using the claims data of the Japanese insurance system.

## Materials and methods

### Study design

This was a retrospective cross-sectional validation study that reviewed health insurance claims data and medical records. Patients who met the inclusion criteria and those who did not were randomly selected from the claims data of patients who visited Kitasato University Kitasato Institute Hospital (Tokyo, Japan) and filed for insurance reimbursement. The medical records of these patients were reviewed to evaluate the validity of the inclusion criteria. Case selection, random sampling, and statistical analyses were conducted in collaboration with the Japan Medical Data Center (JMDC) Corporation (Tokyo, Japan). The flow of the review process is shown in Fig 1.

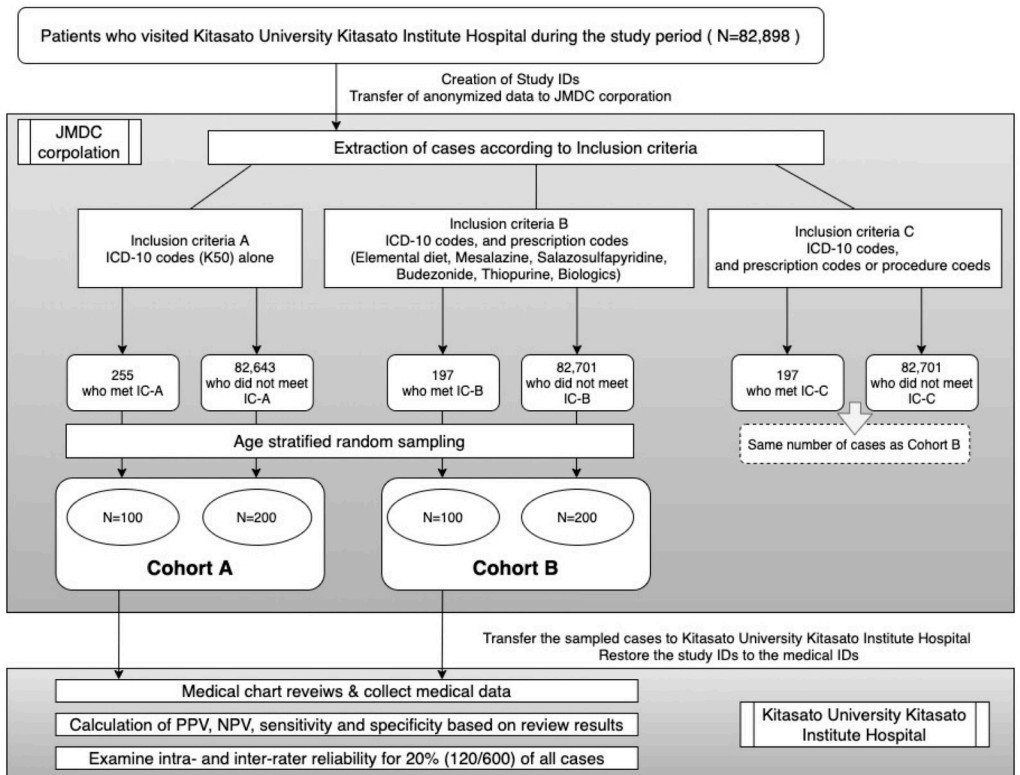

**Fig 1. Study design and data flow, cohort setting.** A total of 82,898 patients were enrolled during the study period. 255 and 197 patients who met the IC-A and B were extracted, respectively. Patients selected for IC-C (n = 197) were excluded from the subsequent analyses due to the same number of patients as in IC-B. JMDC; Japan Medical Data Center, ICD-10; Tenth Revision of the International Statistical Classification of Diseases and Related Health Problems, IC; Inclusion criteria, PPV; positive predictive value, NPV; negative predictive value.

## Japanese health administrative data

Japan has a universal health insurance system, which covers almost all citizens, as they are obliged to join one of the systems according to their occupation and age [20]. At the end of each month, each medical provider files a set of reimbursement invoices to the insurance payer via the review organization. For this reason, medical institutions register all processes, drugs, procedures, and devices that are subject to reimbursement according to the Ministry of Health, Labor and Welfare's standard codes, and this registration information is managed as the Japanese claims database [20, 21].

## Setting

Kitasato University Kitasato Institute Hospital (Tokyo, Japan) is affiliated with Kitasato University; it has the Center for Advanced IBD Research and Treatment, which has 329 hospital beds. It received 865 outpatients and received 163 inpatients per day in FY2019.

## Inclusion criteria

Patients who visited the hospital between January 2015 and December 2019 were included in the first sampling of the claims data. The observation period was set as the maximum period for which insurance claims data were available at the study site. According to the inclusion

**Table 1. Inclusion criteria, and details of the confirmed diagnosis.**

| Criteria | | |
|---|---|---|
| Inclusion Criteria | A | Patients with a confirmed ICD-10 diagnostic code of CD (K50), without a confirmed ICD-10 diagnostic code of ulcerative colitis (K51) or Behcet's disease (M35) in the same month. |
| | B | A + Prescription codes for CD in the same month |
| | C | A + Prescription or Surgical codes for CD in the same month |
| Details of the confirmed diagnosis | a | Confirmed diagnosis at own institution |
| | b | Diagnosed by an IBD specialist or gastroenterologist in another hospital |
| | c | Diagnosed by a primary care physician (with a description of the findings supporting the diagnosis) |
| | d | Diagnosed by a primary care physician (without a description of the findings supporting the diagnosis) |

IBD; Inflammatory bowel disease, ICD-10; Tenth Revision of the International Statistical Classification of Diseases and Related Health Problems, CD; Crohn's disease

criteria listed below, the cases were divided into those that met the inclusion criteria and those that did not (Table 1). Age-stratified random sampling of 100 cases each was performed in cases that met the inclusion criteria and 200 cases from those that did not. The cases extracted using each inclusion criterion (IC-A/B/C) were defined as cohorts (Cohort-A/B/C).

Inclusion criteria A (IC-A; diagnostic code alone): Patients with a confirmed ICD-10 diagnostic code of CD (K50) (S1 Table) but without a confirmed ICD-10 diagnostic code of ulcerative colitis or Behçet's disease in the same month.

Inclusion criteria B (IC-B; diagnostic and prescription codes): Cases fulfilling IC-A and with prescription codes (S2 Table) in the same month as the diagnostic codes.

Inclusion criteria C (IC-C; diagnostic, prescription, and surgical codes): Cases fulfilling IC-A with prescription codes or surgical codes (S2 Table) in the same month as the diagnostic codes.

## Reviewing process

A medical chart review was independently performed by two gastroenterologists (chart reviewers with at least 5 years of clinical experience and training in IBD practice at a specialist center who are engaged in Kitasato University Kitasato Institute Hospital). The reviewers classified cases into three categories based on the gold standard according to the definition by the national guidelines [1] described in the section below as confirmed diagnosis, suspected diagnosis, and negative. If the two reviewers had different diagnoses, the final decision was made after (1) discussion between the two reviewers or (2) consultation with a third reviewer (a gastroenterologist and IBD specialist).

## Gold standard and data collection of clinical information

The following data were collected for each randomly sampled case at the time when the inclusion criteria were met: age, sex, age of onset, disease type (Montreal classification), previous surgery (intestinal/anal), medications for CD, laboratory findings, examination results (upper and lower endoscopy, histopathology, small bowel radiography, magnetic resonance enterography, and intestinal ultrasound findings), discharge summary, referral letter, and registration of intractable disease application. The gold standard was based on the national guidelines of the Japanese Society of gastroenterology [1]. The details of cases with confirmed diagnoses

were categorized as follows: a) diagnosed or confirmed the diagnosis at our own institution, b) diagnosed only by an IBD specialist or gastroenterologist in another hospital; c) diagnosed only by a primary care physician (with a description of the findings supporting the diagnosis), and d) diagnosed only by a primary care physician (without a description of the findings supporting the diagnosis).

## Assessment of validity

For each inclusion criterion, validity was assessed for confirmed and suspected diagnoses. A 2 × 2 contingency table was created, and the validity was mainly calculated by the positive predictive value (PPV). The sensitivity, specificity, and negative predictive value (NPV) were also calculated. A total of 20% (120/600) of the total cases were independently reviewed by two chart reviewers per case to examine inter-rater reliability and another 20% of the total cases were reviewed twice by one chart reviewer with a two-week interval, to examine the intra-rater reliability.

## Statistical analysis

All statistical analyses were performed using STATA/S v. 15.1 (Stata Corporation, College Station, Texas, USA). Continuous variables were expressed as the median interquartile range (IQR) or mean standard deviation (SD). Categorical variables were expressed as integers and percentages (%). A 2 × 2 contingency table was created to calculate the sensitivity, specificity, PPV, and NPV. Inter- and intra-rater reliability was assessed using kappa, weighted kappa, and AC1.

The sample size was set at 100 for cases that met the inclusion criteria and 200 for cases that did not. If the 95% confidence interval for PPV was set to within ±0.1, the required number of cases that met the inclusion criteria was 100. Since the prevalence of CD is 55.6/100,000 in Japan [7], approximately 370,000 cases that did not meet the inclusion criteria were required to detect the exact sensitivity and specificity. However, to ensure feasibility, only 200 cases were selected.

## Ethical considerations

The study was conducted in accordance with the Declaration of Helsinki and Good Clinical Practice guidelines. The Research Ethics Committee of Kitasato University Kitasato Institute Hospital approved the study protocol and all necessary documents (approval number: 19047). The study used data already recorded, and the ethics committee approved a waiver of informed consent.

# Results

## Case extraction and medical record review

A total of 82,898 patients who visited Kitasato University Kitasato Institute Hospital during the study period were enrolled, and 255 and 197 cases who met IC-A and B respectively, were extracted. Although 197 cases were selected for IC-C, they were excluded from later analyses because the number of cases that met IC-C was the same as IC-B (Fig 1). In Cohort-A, PPV was 83.0% for only confirmed diagnosis and 90.0% for confirmed and suspected diagnosis, and in Cohort-B, PPV was 97.0% for only confirmed diagnosis and 100.0% for confirmed and suspected diagnosis (Table 2) (The 2×2 tables are shown in S3 Table).

In Cohort-A, the positive predictive value (PPV) was 0.830 for confirmed and 0.900 for confirmed and suspected Crohn's disease (CD) cases. In Cohort-B, the PPV was 0.970 for

**Table 2. Assessment of validity for each cohort.**

| Cohort | Diagnosis | TP | TN | FP | FN | Sensitivity (95% CI) | Specificity (95% CI) | PPV (95% CI) | NPV (95% CI) |
|---|---|---|---|---|---|---|---|---|---|
| A | Confirmed | 83 | 200 | 17 | 0 | 1.000(0.957–1.000) | 0.922(0.878–0.954) | 0.830(0.742–0.898) | 1.000(0.982–1.000) |
|  | Confirmed & suspected | 90 | 200 | 10 | 0 | 1.000(0.960–1.000) | 0.952(0.914–0.977) | 0.900(0.824–0.951) | 1.000(0.982–1.000) |
| B | Confirmed | 97 | 200 | 3 | 0 | 1.000(0.963–1.000) | 0.985(0.957–0.997) | 0.970(0.915–0.994) | 1.000(0.982–1.000) |
|  | Confirmed & suspected | 100 | 200 | 0 | 0 | 1.000(0.964–1.000) | 1.000(0.982–1.000) | 1.000(0.964–1.000) | 1.000(0.982–1.000) |

*CD; Crohn's disease, TP; true-positive, TN; true-negative, FP; false-positive, FN; false-negative, PPV; positive predictive value, NPV; negative predictive value

confirmed and 1.000 for confirmed and suspected CD cases. The negative predictive value (NPV) is 1.000 because there are no false-negative cases.

The characteristics of the patients who were diagnosed as confirmed and suspected cases in each cohort are shown in Table 3. In Cohort-A, 90 CD patients were diagnosed as confirmed and suspected cases [mean age 43.7±14.0, 62 males (68.9%)]; in Cohort-B (n = 100), the mean age was 44.3±14.7 and included 71 males (71.0%). In Cohort-A, 62% of the patients had CD confirmed by our medical records, 20% by an IBD specialist or gastroenterologist in another hospital, and 1% by primary care physicians without a description of the findings supporting the diagnosis (Fig 2). Of those, 7% were considered to have suspicious diagnoses and 10% were declared negative for CD, based on our medical records. Cases that were declared negative for CD included infectious enterocolitis (n = 4), intestinal Behçet's disease (n = 2), drug-induced enterocolitis (n = 1), intestinal tuberculosis (n = 1), unspecified intestinal stenosis (n = 1), and cirrhosis (n = 1). In Cohort-B, 74% of the patients had CD confirmed by our medical records, 23% by an IBD specialist or gastroenterologist in another hospital, and 3% were considered to have suspicious diagnoses. No cases were declared negative for CD in Cohort-B.

## Inter- and intra-rater reliability

The inter- and intra-rater reliability are shown in Table 4. The weighted kappa coefficient of inter-rater reliability was 0.9903 and that of intra-rater reliability was 0.9948, suggesting that the diagnoses derived by medical record review were valid.

## Discussion

In this study, we first developed algorithms to extract CD cases from the Japanese claims database by assessing the accuracy of claim codes validated by medical chart review.

For a disease with a low prevalence of CD, it is difficult to secure a sufficient number of cases from a single center. Murakami et al. reviewed the number of CD cases from various facilities, and the maximum number of cases at a single specialist center was approximately 320 [7], which is a small number of cases when compared to the 70,700 cases in Japan as a whole [22]. Another issue is that large-scale observational studies are usually conducted in specialist centers, including numerous non-specialized facilities, and may not reflect real-world practice. A large-scale study utilizing big data is therefore necessary to examine populations representing real-world practice. The insurance claims database is a useful resource and has been used in several important studies [23, 24].

Since Japan has a universal health insurance system and almost all citizens are enrolled, the Japanese claims database is a very useful resource for real-world data in database studies. In addition to the databases owned by the government (National Database), commercial databases from private companies are also available (JMDC, Medical Data Vision), which are under contract to different insurance payers, and which are used to conduct database research

**Table 3. Baseline characteristics of each cohort.**

| | Cohort A (N = 90) | Cohort B (N = 100) |
|---|---|---|
| Age (mean ± SD, years) | 43.65±13.99 | 44.33±14.66 |
| Male, n (%) | 62, 68.9% | 71, 71.0% |
| Age at diagnosis (mean ± SD, years) | 28.08±12.48 | 28.64±12.69 |
| Disease duration, (mean ± SD, years) | 10.52±9.13 | 11.46±10.80 |
| Montreal Age at diagnosis, n (%) | | |
| A1 (<16 years) | 6 (6.7%) | 4 (4.0%) |
| A2 (17–40 years) | 68 (75.6%) | 76 (76%) |
| A3 (>40 years) | 13 (14.4%) | 18(18%) |
| unknown | 3 (3.3%) | 2 (2%) |
| Montreal Location, n (%) | | |
| L1 (ileal) | 19 (21.1%) | 21 (21.0%) |
| L2 (Colonic) | 18 (20.0%) | 15 (15.0%) |
| L3 (ileo-colonic) | 49 (54.4%) | 63 (63.0%) |
| + isolated L4 (upper) | 7 (7.8%) | 11 (11.0%) |
| unknown | 2 (2.2%) | 1 (1.0%) |
| Montreal Behavior, n (%) | | |
| B1 (Non-stricturing, non-penetrating) | 46 (51.1%) | 43 (43.0%) |
| B2 (Stricturing) | 29 (32.2%) | 27 (27.0%) |
| B3 (Penetrating) | 13 (14.4%) | 39 (30.0%) |
| + perianal disease | 29 (32.2%) | 43 (43.0%) |
| unknown | 2 (2.2%) | 0 (0.0%) |
| Prior history of surgery, n (%) | | |
| intestine | 25 (27.7%) | 36 (36.0%) |
| peri-anal | 13 (14.4%) | 21 (21.0%) |
| Intractable disease registration, n (%) | 72 (80.0%) | 92 (92.0%) |
| Review result, (confirmed / suspected) | 83/7 | 97/3 |
| Treatment | | |
| Mesalazine, n (%) | 71 (78.9%) | 80 (80.0%) |
| Immunomodulator, n (%) | 44 (48.9%) | 51 (51.0%) |
| Elemental diet, n (%) | 22 (24.4%) | 26 (26.0%) |
| Corticosteroid, n (%) | 12 (13.3%) | 13 (13.0%) |
| Prednisolone, n (%) | 6 (6.7%) | 6 (6.0%) |
| Budesonide, n (%) | 6 (6.7%) | 7 (7.0%) |
| Biologics, n (%) | 37 (41.1%) | 46 (46.0%) |
| Infliximab, n (%) | 20 (22.2%) | 27 (27.0%) |
| Infliximab BS, n (%) | 1 (1.1%) | 1 (1.0%) |
| Adalimumab, n (%) | 16 (17.8%) | 18 (18.0%) |
| Ustekinumab, n (%) | 1 (1.1%) | 1 (1.0%) |
| Vedolizumab, n (%) | 0 (0.0%) | 0 (0.0%) |

*SD; standard deviation.

and to support hospital management by analyzing medical costs. The National Database is a public database that contains data supporting more than 1 billion claims, as well as data and information on specific legal health checkups and guidance [20]. The Diagnosis Procedure Combination (DPC) database holds medical information of inpatients from 1,730 DPC-registered hospitals captured in 2018. The JMDC database is a commercial database that contains

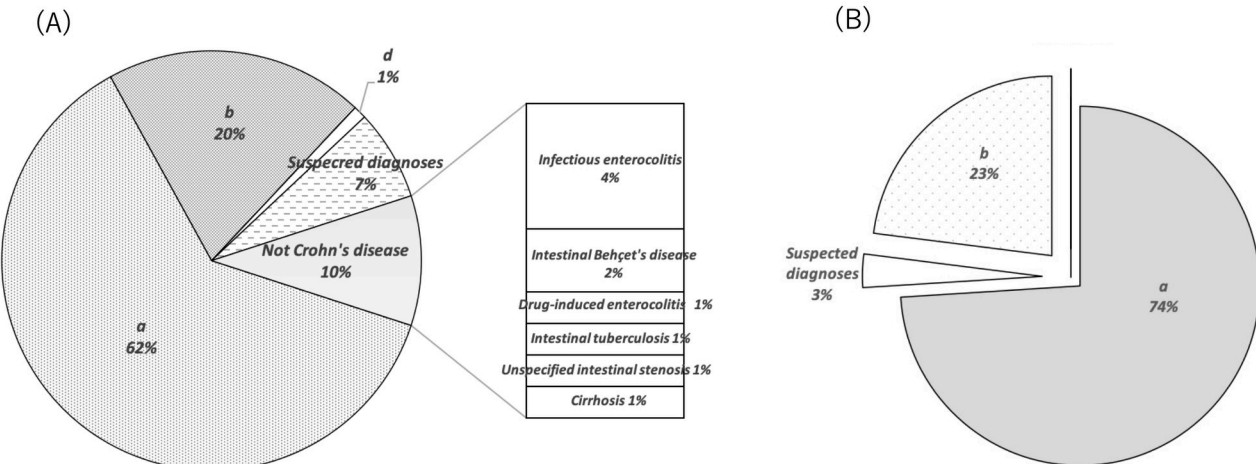

**Fig 2. Details of medical chart review.** (A) Details of medical chart review for Cohort-A. 83% of cases were confirmed for Crohn's disease (CD) (confirmed diagnosis at own institution, or another hospital). 7% were considered suspected diagnosis. Cases denied for CD (10%) included infectious enterocolitis (n = 4), intestinal Behçet's disease (n = 2), drug-induced enterocolitis (n = 1), intestinal tuberculosis (n = 1), unspecified intestinal stenosis (n = 1), and cirrhosis (n = 1). (B) Details of medical chart review for Cohort-B. 97% of cases were confirmed CD. 3% were considered suspected diagnoses. None of the cases were denied CD. *a; Confirmed diagnosis at own institution, b; Diagnosed by an IBD specialist or gastroenterologist in another hospital, c; Diagnosed by a primary care physician (with a description of the findings supporting the diagnosis), d; Diagnosed by a primary care physician (without a description of the findings supporting the diagnosis), CD; Crohn's disease.

claims data for up to 7.3 million insured individuals, which represents approximately 6.1% of the Japanese population between 2005 and April 2020 and includes some salaried employees and their families. A previous study extracted 150 CD cases treated with biologic agents from this database [15]. The Medical Data Vision database is a commercial database that contains data on about 29.8 million patients who received treatment from approximately 400 DPC hospitals in Japan between April 2008 and October 2019. According to a previous database study, about 75,000 CD and ulcerative colitis cases were registered [25]. The validation of our present study is based on claims filed from the medical provider independently of payers; therefore, it is expected to be applicable in any of the claims databases.

There have been many studies on other diseases using the various databases mentioned above. Some utilized prior validation studies [26, 27], but others did not [11, 28]. However, it is possible that a lack of validated algorithms may significantly reduce the reliability of each database study. It is, therefore, extremely important to develop a validated algorithm to extract target diseases from the relevant databases [29].

In fact, in a previous study, the PPV was often remarkably low (60%) for extractions with only a single disease code, while an acceptable PPV (82–91%) was achieved by using repeated detection of the disease code as the extraction protocol [30]. In this study, we found that extraction by diagnostic codes alone (Cohort-A) resulted in the inclusion of other diseases, such as infectious enteritis and Behçet's disease, which suggests that extraction from claims data by ICD-10 code alone is not sufficient. The PPV of confirmed CD cases Cohort-A of this study was 83%. In general, other studies have set the target PPV as 85% or higher [31]. These

**Table 4. Inter- and intra-rater reliability of the medical chart review.**

|  | Kappa (95% CI) | Weighted Kappa (95% CI) | Gwet's AC1 (95% CI) |
|---|---|---|---|
| Inter-rater reliability | 0.9634(0.9136–1.0000) | 0.9903(0.9768–1.0000) | 0.9784(0.9481–1.0000) |
| Intra-rater reliability | 0.9816(0.9457–1.0000) | 0.9948(0.9845–1.0000) | 0.9892(0.9678–1.0000) |

PPV values in Cohort-A did not reach this level. However, Cohort-B, in which prescription codes were added, resulted in a remarkably improved PPV of 97.0%. This is comparable to the PPVs for other diseases in Japan [19, 32, 33] and is therefore considered to be acceptable for general extraction algorithms. The number of cases extracted by IC-B and IC-C, which had additional surgical codes was the same (n = 197). In other words, most cases that underwent a CD-related procedure or surgery were likely to receive the prescription code for CD at the same time, showing that there was little significance in adding the procedure or surgery code.

Some algorithms have been used to extract IBD from other claims databases, such as the algorithm for the Korean National database, which achieved a PPV of approximately 98% by combining the ICD-10 codes, treatment with the incurable disease application code, and the number of hospital visits for IBD [34]. CD is one of the diseases included in the Intractable Disease Registry by the Ministry of Health, Labor and Welfare. However, a certain proportion of patients (20.0% of cohort A and 8.0% of cohort B) were not registered in the registry. This means that Intractable Disease Registry may not necessarily reflect the real world.

Other algorithms that combine Ninth Revision of the International Statistical Classification of Diseases and Related Health Problems (ICD-9) codes with the number of visits and hospitalizations have also reported good PPVs: 81–91% from the Veterans Affairs Health Care System and 94–98% from the Canadian claims database. Ananthakrishnan et al. [35] also reported a PPV of 98% by combining the ICD-9 codes, medical record information, and the complications of IBD for claims data from two tertiary referral hospitals. The results of this study are also comparable to those of other such studies.

We confirmed the extracted patient population in two additional ways. Inter- and intra-rater agreements of the chart review results were confirmed to ensure the reliability of the validation. In addition, the validated cohort in our study was similar to the characteristics of patients in terms of the sex ratio, Montreal classification, prior history of surgery, and previously reported treatment from other specialist centers [35–38].

This study has several limitations. First, although the algorithm developed in this study successfully demonstrated excellent PPV, it is important to note that the study was conducted at a single specialist center, where the prior probability of CD patients among all patients is likely to be much higher than that in the non-specialist centers. Therefore, it is possible that PPV was overestimated compared to real-world clinical practice. In addition, it is also unclear whether cases extracted from the claims database using our algorithm would represent real-world practice in the entire patient population. Further studies to validate our algorithm are warranted from various types of facilities, including non-specialist general hospitals and private clinics.

Second, it is possible that IC-B may inappropriately exclude the true CD patients who have stopped medications and are no longer prescribed. Using the PPV in this study, the numbers of patients who met the IC–A and B before random sampling were estimated to be 211 (PPV 83%) and 191 (PPV 97%), respectively. In other words, IC-B might have excluded approximately 20 patients who had no prescription for several years. If the aim of the study requires to extract such patients together, IC-A should be used with caution to its low PPV. However, considering the disease behavior of CD, it is very rare that a whole set of treatment is discontinued for several years once it has been prescribed.

Third, although the sensitivity (100%), specificity (92–100%), and NPV (100%) shown in our study were excellent, the sample size considered sufficient to accurately calculate these parameters was calculated as 37,000, considering the actual prevalence of CD (55.6/100,000), and thus our sample size (200 cases) is too small. Therefore, the accuracy of these parameters cannot be assured, and 2 × 2 tables with adjusted weights are also assumptive (S3 Table). However, PPV is generally considered to be the most important to develop the extraction

algorithms. Moreover, the sample size required for the calculation of PPV is reported to be much smaller [39, 40]. Therefore, our algorithm is still likely to help appropriately define CD cases from the large-scale claims database.

In conclusion, this study established an algorithm to extract CD from the Japanese claims database and will be of importance in future large-scale real-world studies using the claims database.

## Supporting information

**S1 Table. ICD-10 diagnostic code.**
(XLSX)

**S2 Table. Prescription codes and surgical codes for this study.**
(XLSX)

**S3 Table. A 2 × 2 contingency tables for inclusion criteria and validation criteria.** The number listed is the actual number of validated cases, and the number in parentheses is the assumed number of cases in the entire Kitasato Research Institute Hospital, calculated based on the prevalence calculated from all cases extracted in this study (82,898).
(XLSX)

**S1 File.**
(DOCX)

## Acknowledgments

The authors are grateful to Hiroki Kiyohara, Yuki Watanabe (Center for Advanced IBD Research and Treatment, Kitasato University Kitasato Institute Hospital), Takashi Tanaka, Katsuhiko Nagai (Japan Medical Data Center Co., Ltd.) for their assistance in this study.

## Author Contributions

**Conceptualization:** Hiromu Morikubo, Taku Kobayashi, Takayoshi Nagahama.

**Data curation:** Hiromu Morikubo, Tomohiro Fukuda, Takayoshi Nagahama.

**Formal analysis:** Hiromu Morikubo, Taku Kobayashi, Tomohiro Fukuda, Takayoshi Nagahama.

**Funding acquisition:** Hiromu Morikubo, Taku Kobayashi.

**Investigation:** Hiromu Morikubo, Taku Kobayashi, Tomohiro Fukuda.

**Methodology:** Hiromu Morikubo, Taku Kobayashi, Takayoshi Nagahama.

**Project administration:** Hiromu Morikubo, Taku Kobayashi, Takayoshi Nagahama.

**Supervision:** Taku Kobayashi, Tadakazu Hisamatsu, Toshifumi Hibi.

**Validation:** Hiromu Morikubo, Taku Kobayashi.

**Writing – original draft:** Hiromu Morikubo, Taku Kobayashi.

**Writing – review & editing:** Hiromu Morikubo, Taku Kobayashi, Tomohiro Fukuda, Takayoshi Nagahama, Tadakazu Hisamatsu, Toshifumi Hibi.

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
