## [Decision Letter · Decision Letter 0]

1 Jul 2021

PONE-D-21-09822

Development of algorithms for identifying patients with Crohn’s disease in the Japanese Health Insurance Claims Database

PLOS ONE

Dear Dr. Kobayashi,

Thank you for submitting your manuscript to PLOS ONE. After careful consideration, we feel that it has merit but does not fully meet PLOS ONE’s publication criteria as it currently stands. Therefore, we invite you to submit a revised version of the manuscript that addresses the points raised during the review process.

We look forward to receiving your revised manuscript.

Kind regards,

Valérie Pittet, PhD

Academic Editor

PLOS ONE

2.Please note that PLOS ONE has guidelines on software sharing (https://journals.plos.org/plosone/s/materials-and-software-sharing#loc-sharing-software). Accordingly, we encourage you to make the code for the algorithm described in your manuscript publicly available, if it has not been published in full previously.

“This work was partly supported by JMDC Inc.. JMDC Inc. helped study design, data collection from claims data. There was no additional external funding received for this study. JMDC Inc. URL https://www.jmdc.co.jp/en/index”

4.Your ethics statement should only appear in the Methods section of your manuscript. If your ethics statement is written in any section besides the Methods, please delete it from any other section.

Additional Editor Comments (if provided):

Reviewers' comments:

Reviewer's Responses to Questions

**Comments to the Author**

1. Is the manuscript technically sound, and do the data support the conclusions?

Reviewer #1: Partly

2. Has the statistical analysis been performed appropriately and rigorously? 

Reviewer #1: No

3. Have the authors made all data underlying the findings in their manuscript fully available?

Reviewer #1: Yes

4. Is the manuscript presented in an intelligible fashion and written in standard English?

Reviewer #1: Yes

5. Review Comments to the Author

Reviewer #1: This research is an attempt to investigate disease identification algorithms and validations for making descriptive statistics and analysis of intractable diseases called Crohn's disease (CD) using claims data. So, it is positioned as a valuable study for analyzing many cases that cannot easily be obtained by RCT. On the other hand, I think some of the findings you have introduced should be summarized and commented in a more detailed manner. Please check the the comments listed below;

(P6, L95) How is this study dealing with cases of visiting the facility or being diagnosed and treated for CD before January 2015 and having continuously visited after 2015? What is the reasons for setting this target period? If there was a case in which the patient was heavily prescribed before 2015 but was not prescribed after January 2015, it seems to be included in inclusion criteria A and not included in inclusion criteria B according to the established protocol of this study. But from the viewpoint of pathological condition, isn't it appropriate to interpret it as a case included in inclusion criteria B? In the case of CD, it is unlikely that the same patient will be given or deleted the disease code of "CD" repeatedly, but as to the prescription, it often happens that doctors change the status of prescription for the same CD patient during the course. The author's view needs to be clarified, since the results will vary greatly depending on the setting of the study period and the interpretation of the medication process, whether or not they fall under inclusion criteria B.

(P8, L124) Although the information of registration of intractable disease application is used for confirming whether the case is really CD or not, it may be possible to judge to some extent by checking the presence or absence of the legal number (first 2 digits) of the public funder number for intractable specific diseases patients in the claims data (listed at "KO" code, f). Have you considered reflecting the presence or absence of a public funder number to formulize the inclusion criteria A?

(P10, L148) The number of cases that do not meet the inclusion criteria is 200, isn't it too small? Why did you choose 200 cases? Because of the smallness of the cases, the legitimacy of the findings that there were no CD patients in cases that did not meet the criteria could be questioned. I think it is necessary to devise something to increase the persuasiveness if you appeal that you have obtained the high sensitivity and high specificity.

(S2 Table) Although the prescription code you introduce (ex, "2399009F1149", "2399009F2030") is an individual drug code, it is not a code used for insurance claims. It looks impossible to grasp the medication status from claims data by using this code directly.

6. PLOS authors have the option to publish the peer review history of their article (what does this mean?). If published, this will include your full peer review and any attached files.

Reviewer #1: No

---

## [Author Response · Author response to Decision Letter 0]

4 Aug 2021

We truly appreciate the positive comments and insightful criticisms of the reviewers, especially because adding an accurate interpretation of the inclusion period has greatly strengthened our study.

Clearly the novelty and timeliness of this work was acknowledged. In a competitive field, we believe that PLOS ONE is the appropriate forum for publication of our algorithm for extracting CD patients from insurance databases and our evaluation of its accuracy. Changes in the revised manuscript are denoted in red text for ease of review. 

Reviewer #1: This research is an attempt to investigate disease identification algorithms and validations for making descriptive statistics and analysis of intractable diseases called Crohn's disease (CD) using claims data. So, it is positioned as a valuable study for analyzing many cases that cannot easily be obtained by RCT. On the other hand, I think some of the findings you have introduced should be summarized and commented in a more detailed manner. Please check the comments listed below;

We truly appreciate the favorable feedback. The responses to each comment have been highlighted. 

(P6, L95) How is this study dealing with cases of visiting the facility or being diagnosed and treated for CD before January 2015 and having continuously visited after 2015? What is the reasons for setting this target period?

I agree with the reviewer. The observation period for this study was set from 2015 to 2019 in order to maximize the dataset to avoid the bias as the reviewer mentioned. In Japan, insurance claims data have to be stored for at least three years, and the data prior to that period may not be available. However, we were fortunate that our facility stored the data since 2015 when we initiated this study in 2019, and this is why we took advantage of maximum study period for this study. Additional explanation has been added to the Method section. 

“The observation period was set as the maximum period for which insurance claims data were available at the study site.” Please confirm (Page 6, Lines 97-98).

If there was a case in which the patient was heavily prescribed before 2015 but was not prescribed after January 2015, it seems to be included in inclusion criteria A and not included in inclusion criteria B according to the established protocol of this study. But from the viewpoint of pathological condition, isn't it appropriate to interpret it as a case included in inclusion criteria B? In the case of CD, it is unlikely that the same patient will be given or deleted the disease code of "CD" repeatedly, but as to the prescription, it often happens that doctors change the status of prescription for the same CD patient during the course. The author's view needs to be clarified, since the results will vary greatly depending on the setting of the study period and the interpretation of the medication process, whether or not they fall under inclusion criteria B.

As the reviewer mentioned, CD who had prescriptions before and did not have any prescriptions during the study period should have been excluded by the inclusion criteria B. Considering the PPV in this study, the number of true CD is estimated as 211 (PPV: 83%) for IC-A and 191 (PPV: 97%) for IC-B, which means that 20 true CD patients may be missed in IC-B. Of these, the number of people who fit IC-B before the study period is unknown. However, considering the disease behavior of Crohn's disease, treatment is rarely discontinued for several years once it has been administered. It is expected that such cases are often very mild CD or not CD at all. Additional explanation has been added to the Discussion section. 

“Second, it is possible that IC-B may inappropriately exclude the true CD patients who have stopped medications and are no longer prescribed. Using the PPV in this study, the numbers of patients who met the IC−A and B before random sampling were estimated to be 211 (PPV 83%) and 191 (PPV 97%), respectively. In other words, IC-B might have excluded approximately 20 patients who had no prescription for several years. If the aim of the study requires to extract such patients together, IC-A should be used with caution to its low PPV. However, considering the disease behavior of CD, it is very rare that a whole set of treatment is discontinued for several years once it has been prescribed.” Please confirm (Page 20, Lines 279-285).

(P8, L124) Although the information of registration of intractable disease application is used for confirming whether the case is really CD or not, it may be possible to judge to some extent by checking the presence or absence of the legal number (first 2 digits) of the public funder number for intractable specific diseases patients in the claims data (listed at "KO" code, f). Have you considered reflecting the presence or absence of a public funder number to formulize the inclusion criteria A?

Thank you for the suggestion. This study looked at the utilization rate of applications for intractable diseases in cases of CD diagnoses (Table 3), but applications for intractable diseases were not completed in a certain proportion of cases (20.0% of cohort A and 8.0 % of cohort B were not registered.). In contrast, it is very unlikely that cases with KO codes do not meet the inclusion criteria B because patients need to meet either disease severity (moderate or severe) or a certain level of annual medical expense. In other words, adding intractable disease codes (KO) to the inclusion criteria and use it instead of IC-B would not increase the accuracy but would miss some CD patients. 

This has been added to the Discussion section. 

“CD is one of the diseases included in the Intractable Disease Registry by the Ministry of Health, Labor and Welfare. However, a certain proportion of patients (20.0% of cohort A and 8.0 % of cohort B) were not registered in the registry. This means that Intractable Disease Registry may not necessarily reflect the real world.” (Page18, Lines255-258)

(P10, L148) The number of cases that do not meet the inclusion criteria is 200, isn't it too small? Why did you choose 200 cases? Because of the smallness of the cases, the legitimacy of the findings that there were no CD patients in cases that did not meet the criteria could be questioned. I think it is necessary to devise something to increase the persuasiveness if you appeal that you have obtained the high sensitivity and high specificity.

Thank you for this important comment. Agreeing with the reviewer, we consider that 200 cases that do not meet the inclusion criteria is too few for an accurate assessment of sensitivity and specificity. However, as stated in the Method section, the required number calculated from the prevalence of Crohn's disease is approximately 370,000, which is not feasible. Since the main purpose of the algorithm in this study is to accurately extract Crohn's disease from insurance claims data, we believe that PPV is the most important factor.

We revised the discussion section.

”the sample size considered sufficient to accurately calculate these parameters was calculated as 37,000, considering the actual prevalence of CD (55.6/100,000), and thus our sample size (200 cases) is too small”

(Page 20, Lines287-289)

(S2 Table) Although the prescription code you introduce (ex, "2399009F1149", "2399009F2030") is an individual drug code, it is not a code used for insurance claims. It looks impossible to grasp the medication status from claims data by using this code directly.

Thank you for this important comment. We apologize that the code list has confused you. We have revised the table to the insurance claims codes instead of the individual drug codes (S2 table).

In summary, on behalf of my co-authors I would like to again thank the Editors and Reviewers for their time and helpful comments. I look forward to further comments and I would be happy to provide any further clarifications.

Most sincerely,

Taku Kobayashi, MD., PhD.

Vice Director and Associate Professor

Center for Advanced IBD Research and Treatment

Kitasato University Kitasato Institute Hospital

5-9-1 Shirokane, Minato-ku, Tokyo 108-8642

+81-3-3444-6161

Email: kobataku@insti.kitasato-u.ac.jp

---

## [Editor Report · Decision Letter 1]

30 Sep 2021

Development of algorithms for identifying patients with Crohn’s disease in the Japanese Health Insurance Claims Database

PONE-D-21-09822R1

Dear Dr. Kobayashi,

We’re pleased to inform you that your manuscript has been judged scientifically suitable for publication and will be formally accepted for publication once it meets all outstanding technical requirements.

Kind regards,

Valérie Pittet, PhD

Academic Editor

PLOS ONE
---

## [Editor Report · Acceptance letter]

5 Oct 2021

PONE-D-21-09822R1 

Development of algorithms for identifying patients with Crohn’s disease in the Japanese Health Insurance Claims Database 

Dear Dr. Kobayashi:

I'm pleased to inform you that your manuscript has been deemed suitable for publication in PLOS ONE. Congratulations! Your manuscript is now with our production department. 

Kind regards, 

on behalf of

PD Dr. Valérie Pittet 

Academic Editor

PLOS ONE